# Pharmacological Therapy of Non-Alcoholic Fatty Liver Disease: What Drugs Are Available Now and Future Perspectives

**DOI:** 10.3390/ijerph16224334

**Published:** 2019-11-07

**Authors:** Grazia Pennisi, Ciro Celsa, Federica Spatola, Marcello Dallio, Alessandro Federico, Salvatore Petta

**Affiliations:** 1Section of Gastroenterology and Hepatology, PROMISE, Policlinico Universitario Paolo Giaccone, Piazza delle Cliniche 2, 90127 Palermo, Italy; celsaciro@gmail.com (C.C.); federicaspatola1991@gmail.com (F.S.); salvatore.petta@unipa.it (S.P.); 2Department of Precision Medicine, University of Campania Luigi Vanvitelli, via Pansini 5, 80131 Naples, Italy; alessandro.federico@unicampania.it

**Keywords:** non-alcoholic fatty liver disease, metabolic therapy, metabolic syndrome

## Abstract

The non-alcoholic fatty liver disease (NAFLD) is rapidly becoming the most common cause of chronic liver disease as well as the first cause of liver transplantation. NAFLD is commonly associated with metabolic syndrome (MetS), and this is the most important reason why it is extremely difficult to treat this disease bearing in mind the enormous amount of interrelationships between the liver and other systems in maintaining the metabolic health. The treatment of NAFLD is a key point to prevent NASH progression to advanced fibrosis, to prevent cirrhosis and to prevent the development of its hepatic complications (such as liver decompensation and HCC) and even extrahepatic one. A part of the well-known healthy effect of diet and physical exercise in this setting it is important to design the correct pharmaceutical strategy in order to antagonize the progression of the disease. In this regard, the current review has the scope to give a panoramic view on the possible pharmacological treatment strategy in NAFLD patients.

## 1. Introduction

Nonalcoholic fatty liver disease (NAFLD) is speedily becoming the most common cause of chronic liver disease, liver cirrhosis, hepatic decompensation, hepatocellular carcinoma (HCC), and liver transplantation (LT) in the world [1].

Indeed, the global prevalence rate is estimated around 25.2% of general population, especially in the Western world, with the lowest rate in Africa [2].

NAFLD is a spectrum of pathological conditions that include nonalcoholic steatosis (NAFL), non-alcoholic steatohepatitis (NASH), and cirrhosis. In NAFL, there is hepatic steatosis without evidence of inflammation, whereas in NASH, hepatic steatosis is associated with hepatic inflammation (with or without fibrosis), and finally evolving to cirrhosis, a major risk factor for end-stage liver failure and/or hepatocellular carcinoma (HCC) [3].

NAFLD is usually associated with metabolic syndrome (MetS), and there is accumulative evidence indicating that NAFLD is an independent risk factor for cardiovascular disease (CVD) in adults [4].

Therefore, the treatment of NAFLD is a key point to prevent NASH progression to advanced fibrosis, to prevent cirrhosis and to prevent the development of its hepatic complications (such as liver decompensation and HCC) and even extrahepatic one.

A considerable amount evidence suggests a strong relationship between NASH and lifestyle modifications, including weight loss, physical exercise, and dietary changes. Weight loss and increased physical activity are associated with sustained improvement in liver enzymes, histology, serum insulin levels, and quality of life in patients with NAFLD. The highest rates of NASH reduction, NASH resolution, and fibrosis regression occurred in patients with weight loss ≥ 10% [5]. Furthermore, these interventions help in managing and minimizing the risk of associated comorbidities, such as CVD.

To date, a lot of drugs have been investigated in phase 2 and 3; however, there are no approved medications for NASH. Several targets have been studied for the pharmacologic therapy of NASH, which include anti-inflammatory, anti-fibrotic and anti-apoptotic factors as well as metabolic regulators and anti-oxidant pathways (Figure 1). A possible explanation of the great difficulty in the development of useful drugs against NAFLD is probably related to the disagreement in the definition and assessment of the studies endpoints due to the not completely known mechanisms surrounding the disease as well as the lack of consistent diagnostic and stadiative methods, except the liver biopsy.

Moreover, there are some differences regarding the predisposition, the natural history, the response to therapy due to the ethnicity of analyzed population, that make the interpretation of the study results a difficult challenge.

## 2. Pharmacological Treatment: Different Drugs for Different Therapeutic Targets

All the knowledge contained in the following section are based on the guidelines for the management of NAFLD.

### 2.1. Insulin Sensitizers and Antioxidants

A large number of studies has already shown how insulin resistance is a trigger which induces hepatic steatosis [3]. Lipid accumulation is due to increased flux of free fatty acids from the adipose tissue driven by insulin resistance, that represents the major risk factor for this condition [6], over-stimulation of de novo lipogenesis depends on hyperinsulinemia, and a relative impairment in mitochondrial beta-oxidation [7]. Thiazolidinediones (TZD), such as rosiglitazone and pioglitazone, through different pathways, modulate adipose tissue distribution, decreasing visceral fat, including hepatic fat. TZD regulate insulin sensitivity by the activation of peroxisome proliferator-activated receptor (PPAR)-γ, that modulates the transcription of genes involved in lipid metabolism [8]. Consistent data suggest their efficacy to improve histological and clinical features of NASH. Rosiglitazone and pioglitazone determinate a significant histologically improvement in lobular inflammation and steatosis, but no significant improvement was showed in terms of liver fibrosis, with the exception of recent study of 2016 by Cusi et al. [9]. In this trial pioglitazone (compared to placebo) was administered with a dosage of 45 mg/daily in patients with prediabetes or type 2 diabetes mellitus (T2DM) for 18 months. Pioglitazone group (58%) achieved primary outcome (NAS impovement ≥ 2 points without worsening of fibrosis) than patients in the placebo group (17%) (*p* < 0.001) [9]. NASH resolution occurred in 51% of pioglitazone-treated patients versus 19% of those receiving placebo (*p* < 0.001) with the overall NAS improving in 66% versus 21%, respectively (*p* < 0.001). The fibrosis score also improved significantly with pioglitazone (*p* = 0.039) and its progression over 18 months occurred in only 12% of pioglitazone-treated patients compared with 28% placebo (*p* = 0.039) [9]. Moreover, this is to date, the only trial where TZD was given for more than one year, so duration of treatment may play a role on histological findings in NASH. Bone fractures in women, weight gain, and, rarely, congestive heart failure are concerning potential long-term side effects. Therefore, TZD could be used in T2DM and non-T2DM patients with biopsy-proven NASH [8].

### 2.2. Vitamin E

Oxidative stress is involved in the progression to NASH and advanced fibrosis in NAFLD patients and previous data have reported a strong relationship between the degree of oxidative stress and severity of NAFLD [10]. Vitamin E has been investigated in the treatment of NASH because of its anti-oxidant effect that can protect cellular structure integrity against injury from lipid peroxidation and oxygen-free radicals. However, epidemiological studies showed that low plasma levels of vitamin E (<26.8 nmol/mL) are not only associated with presence of NASH [11] but also with increased all-cause mortality in NAFLD patients [12]. Vilar-Gomez et al. have shown that vitamin E use was associated with improved clinical outcomes in patients with histological proven NASH and bridging fibrosis or cirrhosis. This retrospective analysis included 90 patients with NASH and bridging fibrosis or cirrhosis that took 800 IU/day of vitamin E for ≥2 years and they was compared by propensity score with 90 controls. After a 5-year median follow-up, Vitamin E users have higher transplant-free survival (78% vs. 49%, *p* < 0.01) and lower rates of hepatic decompensation (37% vs. 62%, *p* = 0.04) than controls. Vitamin E treatment decreased the risk of death or transplant (HR: 0.30, 95% CI: 0.12–0.74, *p* < 0.01) and hepatic decompensation (HR: 0.52, 95% CI: 0.28–0.96, *p* = 0.036). Despite of benefits on hepatic decompensation and survival, vitamin E was not associated with a reduced risk of hepatocellular carcinoma. These advantages were reported in both diabetics as well as non-diabetics patients [13]. In the PIVENS study, vitamin E administered at a dose of 800 IU/D for 96 weeks was compared with pioglitazone 30 mg/D and placebo. Vitamin E treatment, as compared with placebo, was associated with a significantly higher rate of improvement in NASH (43% vs. 19%, *p* = 0.001), but not significant if pioglitazone as compared with placebo (34% and 19%, *p* = 0.04). Vitamin E and pioglitazone were associated with reductions in hepatic steatosis (*p* = 0.005 and *p* < 0.001, respectively) and lobular inflammation (*p* = 0.02 and *p* = 0.004, respectively) but not with improvement in fibrosis scores (*p* = 0.24 and *p* = 0.12 respectively) [14]. Different concerns exist over the long-term use of vitamin E, especially at doses > 400 IU/d, regarding increases in all-cause mortality, prostate cancer, and hemorrhagic stroke [15,16,17]. Based on current evidence and despite the potential long-term risks, vitamin E could be used in non-T2DM patients with biopsy-proven NASH [13].

### 2.3. Silymarin

Silymarin is derived from the milk thistle plant *Silybum marianum* and has been considered for a long time as an herbal remedy for liver diseases. It consists in a mixture of 6 major flavonolignans (silybins A and B, isosilybins A and B, silychristin, and silydianin), and other minor polyphenolic compounds [18]. The antioxidant, anti-inflammatory, and antifibrotic properties of silymarin was demonstrating by several in vitro and animal studies [19,20,21,22] Its functions include: protection against radical-induced damage of cell membrane, simulation of polymerase and RNA transcription and regulation of some cell signaling pathways and biological axes like the insulin receptor substrate 1/phosphatidylinositide 3-kinase/Akt pathway and the nicotinamide adenine dinucleotide/sirtuin 1 axis, respectively [23,24].

In a randomized, double-blind, placebo-controlled trial of consecutive patients with histological-proven NASH, silymarin 700 mg was administered 3 times daily for 48 weeks. It showed a histological and non-invasive (based on liver stiffness measurements) improvement of fibrosis despite placebo group (22.4% and 6% respectively, *p* = 023). NAFLD fibrosis score (NAS) was also reduced of 30% (*p* < 0.001) compared with baseline in silymarin treated group but these changes were not observed in the placebo group [25]. However, further studies are needed before to suggest silymarin use for NASH.

## 3. Drugs in Development

Numerous drugs have been tested for the treatment of NAFLD. Due to the not completely clear pathogenesis of NAFLD, the development is complex and involves different molecules with a variety of targets. Clinically meaningful benefit is the main aim of these drugs with a histologically proven improvement requirement: NASH resolution without worsening of fibrosis or improvement in at least one stage of fibrosis without worsening of NASH [26]. Below, we will discuss on the most relevant classes of drugs currently under investigation (phase 2 and 3).

### 3.1. Farnesoid X receptor (FXR) Agonists

FXR is a nuclear receptor identified in 1995 and activated by farnesol pyro-phosphate [27] and exists as two human variants, FXRα (NR1H4) and FXRβ (NR1H5). Since bile acids (BAs) are the principal FXR ligands [28], it is abundantly expressed in the organs involved in their metabolism and transport, such as liver, intestine, and kidney, but also in adipose tissue and adrenal gland. Upon ligand activation, FXR binds to the transcriptional responsive elements as either monomers or heterodimers with retinoid X receptor (RXR) [29]. Its main function is to set BAs enterohepatic circulation. After feeding, intestinal FXR is activated by BAs, induces BA-binding protein in enterocytes, and induces BA and lipids absorption. Additionally, intestinal FXR activation increases the expression of fibroblast growth factor (FGF)15/19 and, by via portal circulation, binds to the FGF liver receptor CYP7A1, the rate-limiting enzyme of BAs synthesis. When hepatocyte FXR is activated by BAs, it inhibits CYP7A1 expression through up-regulation small heterodimer partner (SHP, NR0B2) and increases the expression of ABCB11 (bile salt export pump, BSEP), the main BAs transporter from hepatocytes to bile canaliculi [30].

Many studies conducted in mice and humans reported that hepatic FXR activation reduced fatty liver [31]. Whereas few genes involved in lipid metabolism are directly regulated by FXR activation, the anti-steatotic effect of FXR may be indirect, probably through improvement of insulin resistance and lipoprotein transport.

The role of intestinal FXR in hepatic steatosis is not clear. Activation of FXR induced FGF15/19 expression in the intestine and leads to a negative feedback of BAs synthesis in the liver by inhibition of CYP7A1 gene expression. Pharmacological administration of FGF15/19 increased the metabolic rate and reversed high fat-induced diabetes, decreasing adiposity [32]. Therefore, a specific modulation of FXR signaling, liver and intestine-specific FXR agonist, intestine-specific FXR antagonist, can be a potential treatment of metabolic disorders, including non-alcoholic fatty liver disease [33].

Obeticholic acid (OCA), also known as INT-747, is 6α-ethyl derivative of chenodeoxycholic acid (CDCA) and represents the first-in-class selective steroid FXR agonist. It is approximately 100-fold potent than CDCA for FXR activation [34].

Since it reduces liver fat and fibrosis in animal models of fatty liver disease, in 2011 a clinical trial was designed to assess the efficacy of obeticholic acid in non-alcoholic steatohepatitis. The farnesoid X receptor ligand obeticholic acid in NASH treatment [35] phase 2b trial was a multicenter, randomized trial, double-blind, of 72 weeks of obeticholic acid versus placebo (PBO) in 283 patients with biopsy-proven non-alcoholic steatohepatitis. It was found that, 45% patients in the OCA group had improved liver histology (2-point or greater improvement in NAFLD activity score without worsening of fibrosis) compared with 21% of patients in the PBO group (*p* = 0.0002). Secondary histological outcomes included changes in the individual scores for hepatocellular ballooning (*p* = 0.03), steatosis (*p* = 0.001), lobular inflammation (*p* = 0.006), portal inflammation (*p* = 0.9), and improvement of fibrosis (*p* = 0.004) in OCA and PBO arm respectively. Despite these improvements in histological features of NASH, the proportion of patients with resolution of non-alcoholic steatohepatitis did not differ in patients who received OCA compared with PBO (*p* = 0.08). Clinical adverse events were generally mild to moderate in severity and were similar in the two groups. An exception was pruritus, reported in 23% of OCA-treated patients and 6% of PBO-treated patients (*p* < 0.0001) [36].

The results of 18 months interim analysis of phase 3 REGENERATE trial were shown during the last International Liver Congress 2019. The population included 931 patients (PBO (*n* = 311), OCA 10 mg (*n* = 312) or OCA 25 mg (*n* = 308)), with hystological NAFLD and fibrosis F2–F3. The improvement of fibrosis with no worsening of NASH was met by 11.9% PBO, 17.6% OCA 10 mg (*p* = 0.0446), and 23.1% OCA 25 mg (*p* = 0.0002) patients. The improvement of NASH without worsening of fibrosis was not statistically significant. Itch was the most common adverse event and forced the discontinuation especially in OCA 25 mg arm. Increases in LDL were also observed by week 4 [37].

NGM282 (known as M70) is a non-tumorigenic, engineered variant of fibroblast growth factor (FGF)-19 that binds FGF receptors 1c and 4 to reduce lipotoxicity due to lipid accumulation. Rodent FGF15 and the human analog FGF19, expressed in enterocytes and secreted into portal circulation, have a FXR-dependent transcription. Their functions are to regulate bile acid synthesis and carbohydrate metabolism. FGF15/19 represses transcription of cholesterol 7α-hydroxylase, inhibiting bile acid synthesis in the liver and stimulating the gallbladder to fill with bile. FGF15/19 stimulates FGFR4/β-klotho receptor complex in the liver to promote hepatic glycogen and protein synthesis and suppresses hepatic gluconeogenesis. However, FGF15/19 is implicated in liver tumorigenesis through liver proliferation [38]. NGM282 is a modified analog of FGF19 designed to be a nontumorigenic variant but maintaining metabolic functions of native FGF19 [39]. It was studied in a randomized, double-blind, placebo-controlled phase 2 trial (NCT02443116) in 82 patients with histological-proven NASH with ≥8% absolute liver fat content by MRI proton density fat fraction (MRI-PDFF). It was administrated subcutaneous, once-daily, for 12 weeks. It induced marked relative reduction from baseline in steatosis with a 57% and 45% (*p* < 0.0001) at the 6- and 3-mg doses, respectively. Improvements in ALT (*p* < 0.0001), AST (*p* < 0.0001) were also observed, but not in glycemic control. Injection site reactions were reported in 54% and 41% of subjects in the 6- and 3-mg dose groups relative to 7% in the PBO group. Others adverse events are: diarrhea were also higher in treated subjects (36% and 41% at the 6- and 3-mg dose levels, respectively) relative to PBO (22%), nausea, abdominal pain, and distension [40]. In an exploratory, non–placebo-controlled, 12-week study, 67% (1 mg) and 74% (3 mg) of subjects showed an improvement in steatosis, 33% (1 mg), and 42% (3 mg) showed improvement in inflammation, 42% (1 mg) and 53% (3 mg) showed improvement in ballooning, and 25% (1 mg) and 42% (3 mg) showed improvement in fibrosis, as assessed by liver histology [41].

Cilofexor, also known GS-9674, is a non-steroidal FXR agonist. First data were reported from a phase II study with 140 NASH patients, treated with cilofexor at 100 or 30 mg or placebo orally, once daily for 24 weeks. A significant decrease in hepatic steatosis of at least 30% assessed by MRI-PDFF was observed in 38.9% of patients treated with cilofexor 100 mg (*p* = 0.011), 14% treated with cilofexor 30 mg (*p* = 0.87), and 12.5% treated with placebo. Further significant observations in the cilofexor-treated patients were improvements in serum gamma-glutamyltransferase (GGT) and serum 7α-hydroxy-4-cholesten-3-one (C4). Cilofexor not induces changes in lipid profile (such as OCA) and was generally well tolerated, except for moderate to severe pruritus occurred in 14% of patients in the cilofexor 100 mg arm and in 4% in the cilofexor 30 mg and PBO arms [42]. A Phase II study [43] is investigating treatment with cilofexor alone or in combination with Firsocostat (GS-0976), an acetyl-CoA carboxylase (ACC) inhibitor, and Selonsertib, an apoptosis signal-regulating kinase 1 (ASK-1) inhibitor, in patients with advanced fibrosis due to NASH. This randomized, double-blind 52-week trial will evaluate improvement in fibrosis without worsening of NASH, adverse events, and plasma laboratory changes. First data from a proof-of-concept study using cilofexor were presented at the International Liver Congress 2018 [44].

Tropifexor is a non-steroideal FXR agonist and its safety, tolerability and efficacy were studied in a phase 2 clinical trial (FLIGHT-FXR). At 12 weeks, a significant decrease in hepatic steatosis of at least 5% assessed by MRI-PDFF was observed in 33.3% of patients treated with Tropifexor 90 μg, 27.8% treated with Tropifexor 60 μg, and 14.6% treated with PBO. Furthermore, a dose–response decrease in GGT levels was observed as well as increases in FGF19. Adverse events, especially pruritus and increase in LDLc and HDLc, were reported in 90 μg Tropifexor arm [45].

Tropifexor is also under investigation in a randomized, double-blind combination trial (TANDEM) with Cenicriviroc, an C-C chemokine receptor-type (CCR) 2/5 inhibitor. This combination therapy should hit all pathways involved in NASH pathogenesis (metabolic, anti-inflammatory, and anti-fibrotic). The 48-week trial will assess improvement in liver histology and occurrence of adverse events in approximately 200 patients with NASH and liver fibrosis [46].

However, the available evidence from clinical trials with nonsteroidal FXR agonists suggests that these class of drugs may hold the potential to develop forth better FXR-targeting molecules with improved pharmacological actions and reduced adverse effects especially in terms of cholesterol metabolism and itch.

### 3.2. Peroxisome proliferator-activated receptors (PPAR) Agonists

PPAR are ligand activated transcription factors and they represent a subfamily of the NR1C nuclear receptors, including three isotypes: PPARα (NR1C1), PPARδ (also called PPARβ or NR1C2), and PPARγ (NR1C3). Each isotype has different tissue distribution patterns and functions, playing a key role in glucose and lipid metabolisms and inflammation [47]. PPARγ and its downstream effector enzymes (i.e., acetyl-CoA carboxylase (ACC) and fatty acid synthase (FAS)) regulate gene expression in liver, vascular endothelium, adipose, and muscle tissues and they were studied first [48]. The effects of pharmacological activation of PPARγ through the administration of pioglitazone (see above) are mediated by the differentiation of pre-adipocytes in adipocytes, that leads to the anatomic redistribution of triglycerides from ectopic sites (liver and muscle) to adipose tissue [49]. It has been showed that PPARγ activation increases fatty acids β-oxidation and adiponectin levels in NASH patients, reducing the hepatic supply of fatty acids from adipose tissue [50,51,52]. It is able to dynamically respond to feeding or starvation modulating the transcription of genes involved in metabolism homeostasis [53,54]. Furthermore, PPARα downregulates signaling pathways involved in inflammation and acute phase response in rodent models [55]. Fibrates act as PPARα agonists and they has been evaluated for NAFLD treatment: it has been showed a positive effect on transaminase and GGT, but no significant changes were observed in histologically assessed steatosis, inflammation, and fibrosis [56,57,58]. It has been hypothesized that the weak effect of PPARα agonists in human could be related to the lower expression of PPARα in human liver compared to mouse or to an inverse correlation between progressive stages of liver fibrosis and hepatic PPARα [54,59]. PPARδ activation improves insulin sensitivity reducing hepatic glucose output and increasing HDL levels and it acts on Kupffer cells and macrophages inducing anti-inflammatory effects [60,61]. Some of these positive metabolic effects were also observed in humans, using GW501516, a PPARδ agonist that was compared with a PPARα GW590735 in a pilot RCT: GW501516 significantly reduced triglicerydes, apolipoprotein B, LDL cholesterol, insulin, and liver fat content [62].

Elafibranor (GFT505) is an agonist of both PPARα and PPARδ, that was initially evaluated in animal models, showing protective effects on steatosis, inflammation and fibrosis. The effects of Elafibranor on liver seem to be mediated not only by PPAR-α-dependent, but also by PPAR-α-independent mechanisms [63]. Benefits in terms of improvement of insulin-resistance and lipid panels were also observed in humans with dyslipidemia, prediabetes and over diabetes in phase II trials [64,65]. Elafibranor was assessed in a multicenter international RCT (GOLDEN-505), including patients with histologic diagnosis of NASH without cirrhosis [66]. Two dosages (80 mg or 120 mg once a day) were compared against placebo over 52 weeks. In the intention to treat analysis, 92 patients received placebo and 93 and 89 patients received Elafibranor 80 and 120 mg, respectively. Two primary outcomes were considered: the first, namely a score of 0 of at least 1 between steatosis, ballooning, and inflammation (reversal of NASH) without worsening of fibrosis (progression of patients with fibrosis stage ≤ 2 to F3-F4 or from F3 to F4) was established at the beginning of the trial, whereas the second, more stringent, was adopted after the study was completed [67]. no differences were observed between patients treated with Elafibranor and placebo (OR 1.53, 95% CI 0.70−3.34, *p* = 0.280); conversely, using the modified definition, the response rate was significantly higher in those who received Elafibranor 120 mg in comparison with placebo (OR 2.31, 95% CI 1.02−5.24, *p* = 0.045), with a lower placebo effect. A post-hoc analysis restricted to 234 patients with NAS score ≥ 4 at baseline showed that treatment with Elafibranor 120 mg (*n* = 75) was significantly associated with higher response rate in comparison with placebo (*n* = 76), using both definitions of response, although it should be considered that no stratification according this covariate was performed at randomization. An international phase 3 trial (RESOLVE-IT, NCT02704403) is ongoing: the recruitment of patients needed to assess the impact of Elafibranor on primary histological outcome (resolution of NASH without worsening of fibrosis after 72 weeks of treatment) has been completed, but it will continue to evaluate also survival and liver-related outcomes.

IVA337 (Lanifibranor) is a pan-PPAR agonist that acts moderately and in a well-balanced way on the three PPAR isoforms. Interesting results have been observed in several preclinical models. Anti-fibrotic effects, especially the inhibition of platelet-derived growth factor (PDGF)-induced proliferation and stiffness-induced activation of hepatic stellate cells (HSCs) were showed in vitro. When administered to animal models, IVA337 showed positive effects in reduction of body weight, improvement of insulin resistance, prevention of steatohepatitis, decrease of steatosis, ballooning, inflammation, and reduction of profibrotic and proinflammatory gene expression. Interestingly, IVA337 demonstrated antifibrotic efficacy in the CCl4-induced liver fibrosis model in terms of prevention or reversion of fibrosis [68]. Therefore, it has been speculated that the combined activation of the three PPAR isoforms could be better than specific or dual agonists, through the interaction with different fibrosis pathways in NASH. For these reasons, IVA337 represents a promising candidate for NASH treatment.

### 3.3. Inhibitors of De Novo Lipogenesis

Pharmacological treatments have de novo lipogenesis and steatosis as main targets and include inhibitors of acetyl-coA carboxylase (ACC) and aramchol. ACC enzymes (ACC1 and ACC2) are involved in fatty acid synthesis and oxidation through the conversion of Acetyl-coA to Malonyl-CoA, that is the rate-determining step in de novo lipogenesis. The pharmacologic inhibition of ACC was assessed in diet-induced rodent models of NAFLD, showing a significant effect in decreasing hepatic malonyl-CoA levels, hepatic de novo lipogenesis and hepatic insulin resistance with an increase in hepatic ketogenesis [69]. However, inhibition of ACC has been associated with significantly increased plasma triglycerides levels, and this is probably mediated by the increase in hepatic production of very low-density lipoprotein production and by a decrease of the effects of lipoprotein lipase on the clearance of triglycerides. Interestingly, ACC inhibition could have anti-neoplastic effects: it has been showed that mutations of the AMP-activated protein kinase (AMPK) phosphorylation sites on ACC1 increase de novo lipogenesis and stimulate the proliferation of human HCC cells. The inhibition of ACC1 with ND-654, an agent that mimics the effects of ACC phosphorylation, showed a protective effect, not only on de novo lipogenesis, but also on the development of HCC in mice [70].

In humans, inhibitors of ACC were firstly assessed in a randomized controlled crossover trial conducted in overweight and obese but otherwise healthy male subject, showing a good safety profile and promising dose-dependent results in terms of reduction of de novo hepatic liver lipogenesis in response to fructose administration [71].

After a small open-label pilot study in NASH (*n* = 10), GS-0976, an inhibitor of ACC, was assessed in a phase 2 RCT [72,73]. Ninety-three patients with NASH (diagnosed with magnetic resonance elastography or with histology) were assigned to GS-0976 (46 patients to 20 mg dosage and 47 to 5 mg), while 26 patients received placebo for 12 weeks. In comparison with placebo, GS-0976 20 mg was significantly associated with a relative reduction of at least 30% from baseline in magnetic resonance imaging-estimated proton density fat fraction (MRI-PDFF) (48% vs. 15%, *p* = 0.004). Additionally, median relative decreases in MRI-PDFF were more pronounced for GS-0976 20 mg in comparison with placebo. These outcomes resulted similar between placebo and GS-0976 5 mg. Furthermore, a dose-dependent reduction in tissue inhibitor of metalloproteinase-1 (TIMP-1) was noted in patients treated with higher dosage, although no significant changes were showed in MRI elastography-measured liver stiffness. It was found that 13% of patients taking GS-0976 showed elevation in serum triglycerides, but it often improved spontaneously or responded to standard therapy. Results from future trials of GS-0976, in monotherapy or in combination with other agents, are needed to confirm its promising effects on improvement of steatosis.

Aramchol, a conjugate of cholic and arachidic acid, belongs to the family of synthetic fatty-acid/bile-acid conjugates (FABACs). It acts inhibiting steatoyl-coenzime A desaturase (SCD), an enzyme involved in lipotoxicity and microinflammation through the conversion of saturated fatty acids into monounsaturated fatty acids. It was assessed both in cultured cells and in mice, showing an effect in decreasing liver fat content and plasma cholesterol levels, and in increasing cholesterol efflux from macrophages [74,75]. Furthermore, it could also have antifibrotic effects that take place directly targeting HSCs in downregulation of collagen and alpha-smooth muscle actin [15,76]. A multicenter phase 2 RCT was conducted in 60 Israelian patients with histologically proven NAFLD (NASH was present in only 10%) who were randomized to receive Aramchol (100 or 300 mg) or placebo. Patients treated with Aramachol 300 mg showed a significant reduction in liver fat content, assessed by MR spectroscopy (MRS), after 3 months of treatment. Conversely, the same was not observed in patients treated with Aramchol 100 mg. No significant improvements were observed in transaminase levels and adiponectin. More recently, results from an international multicenter phase IIb RCT were presented during the last AASLD Liver Meeting in 2018 [77]. A group of 247 patients with histologically proven NASH, NAS ≥ 4 without cirrhosis, and type 2 diabetes/prediabetes were randomized to receive Aramchol 400 mg (*n* = 101), Aramchol 600 mg (*n* = 98) and placebo (*n* = 48). Primary endpoint was the decrease from baseline in steatosis assessed by MRS; secondary endpoints were fibrosis score improvement without worsening of NASH and NASH resolution without worsening of fibrosis. Treatment with Aramchol 600 mg was associated with reduction of at least 5% in liver fat from baseline in 47% of patients (versus 24% in placebo, *p* = 0.0279, OR 2.77) (95% CI 1.12–6.89) and with NASH resolution alone in 19% of patients (versus 7.5% in placebo, *p* = 0.046). A marginally significant effect of Aramchol 600 mg was shown in terms of NASH resolution without worsening of fibrosis (16.7% versus 5% in placebo, *p* = 0.051 OR 4.74, 95% CI 0.99–22.7), while a significant effect on fibrosis improvement without worsening of NASH was not observed. Conversely both dosages of Aramchol induced a significant reduction in transaminase levels, especially after treatment with the higher dosage. Safety and tolerability profiles were good, in absence of signals for hepatotoxicity or increase in body weight and lipid parameters. However, these results need a confirmation from phase 3 RCTs.

### 3.4. Agonist of Thyroid Hormone Receptor

Thyroid hormones play a key role in the regulation of homeostasis, lipid and glucidic metabolism, regulation of body weight and adipogenesis through the effects on thyroid hormone receptor β (THR β) that represents also the predominant THR isoform in the liver [78]. It has been showed that thyroid dysfunctions, particularly hypothyroidism, are associated with insulin resistance and dyslipidemia and with cardiovascular mortality. Furthermore, hypothyroidism has been recently shown as an independent risk factor for NAFLD [79,80,81,82,83,84]. For these reasons, THR β agonists, particularly MGL-3196 and VK-2809, are still under investigation.

MGL-3196 (Resmetirom) is a highly selective THR β agonist that showed positive effects on reduction of LDL cholesterol and triglycerides in phase I studies. It was assessed in a multicenter double-blinded RCT versus placebo in patients with NAS score ≥ 4 and fibrosis stage F1-F3 [85]. Dosage of 80 mg once daily was assigned to 78 patients and 38 patients received placebo for 36 weeks. The primary endpoint at 12 weeks (relative reduction in MRI-PDFF) was met (−36% in treated arm versus −9% in placebo arm). Conversely, the histological endpoint at 36 weeks (2-points NAS reduction with ≥1 point decrease in ballooning or inflammation) was not met (51% in treated arm vs. 32% in placebo arm, *p* = 0.09). However, limiting the analysis to patients treated with MGL-3196 who obtained a reduction of at least 30% in MRI-PDFF, response rate was significantly higher (65%, *p* = 0.006) in comparison with placebo, suggesting a relationship between early reduction in liver fat at 12 weeks and NASH histological improvement at 36 weeks. Furthermore, MGL-3196 showed efficacy in achieving NASH resolution (27% vs. 6%, *p* = 0.02) and in decreasing significantly ALT levels, fibrosis biomarkers and lipids, with an acceptable safety profile. A phase III trial has recently started and it is ongoing (NCT03900429).

VK-2809 is another THR β agonist evaluated in a 12-week phase II RCT, in whom 16 patients received VK-2809 10 mg once a day, 16 patients received 10 mg every other day and 15 patients were assigned to placebo [86]. Daily dose administration was associated with a median reduction of 60% (*p* < 0.01) at 12 weeks of liver fat content in comparison with placebo, although an increase in ALT levels was observed, especially during early treatment. However, after 12 weeks of treatment, ALT levels were similar between VK-2809 and placebo. Results from phase II trials that evaluated THR β agonists in NASH seems to be attractive, but further data from larger phase III trials with histological endpoints are needed. At the moment there is not any medication of this category approved for clinical use in this setting.

### 3.5. Antidiabetic Drugs

Insulin resistance is one of the key mechanisms involved in NASH pathogenesis and relationship between NAFLD and type 2 diabetes is well-known. NAFLD at baseline represents an independent predictor of type 2 diabetes and the presence of type 2 diabetes independently predicts the occurrence of NAFLD [87,88]. For these reasons, several antidiabetic pharmacological treatments have been studied in NAFLD patients with and without diabetes [8]. We have focused on the role of the glucagon-like peptide-1 (GLP-1) analogues and the sodium-glucose cotransporter 2 (SGLT-2) inhibitors. The rationale for the use of this category of drugs in this setting derived from the strong association between type 2 diabetes mellitus and NAFLD because of sharing of several pathogenetic mechanisms as well as the reciprocal influences in the worsening of pathological picture [89].

GLP-1 is released from intestinal epithelial L-cells under stimulation of meals and it binds to its receptor, stimulating pancreatic beta cell insulin secretion, inhibiting the release of glucagon and regulating the homeostasis of glucose. GLP-1 analogues lower glucose blood levels, but they also have other pleiotropic central and peripheral extra-pancreatic effects. It has been shown that they are involved in the regulation of the sense of appetite and in delaying gastric emptying. Furthermore, they are able to induce weight loss, to improve cardiac function and to express also effects directly on liver [90]. However, the mechanisms by which GLP-1 analogues act directly on liver, determining a reduction in steatosis, inflammation, and fibrosis are not completely clear, therefore further evidence is needed [91].

Liraglutide is a long-acting GLP-1 analogue licensed not only for type 2 diabetes, but also for the treatment of obesity. A 26-week administration of liraglutide has been shown to be associated with a significant improvement in liver enzymes and with a good safety profile in an individual patient data meta-analysis of six RCTs, concerning more than 4000 diabetic patients [92]. The Liraglutide efficacy and action in NASH (LEAN) trial, that is a multicenter phase II RCT, evaluated the impact of liraglutide on hepatic histology in 52 patients with biopsy-proved NASH [93]. Forty-eight weeks of subcutaneous liraglutide (1.8 mg/day) were compared versus placebo and primary endpoint was NASH resolution without impairment of fibrosis. Thirty-nine percent of patients treated with liraglutide compared to 9% placebo group patient (*p* = 0.019) met the primary endpoint, whereas no significant differences were observed in progression of fibrosis between two groups. Two patients receiving liraglutide (in comparison to 8 patients assigned to placebo group) displayed worsening of fibrosis. Although treatment with liraglutide was associated with a significant improvement in steatosis and ballooning, no significant differences were observed in lobular inflammation and in NAFLD activity score. It has been speculated that the positive effect of liraglutide on hepatic histology may be due to synergistic and multifactorial effects between its direct action on liver and the weight loss. Liraglutide showed a good safety profiles, regardless of the severity of NASH and of the presence of cirrhosis. Interestingly, the metabolic effects of liraglutide was further elucidated in a sub-study of LEAN trial, showing that liraglutide was able to induce a reduction in free fatty acids concentration, peripheral lipolysis, de novo lipogenesis, hepatic neoglucogenesis, and pro-inflammatory cytokines, whereas it had a positive effect in the improvement of insulin resistance and in increasing adiponectin levels [94]. More recently, a pilot RCT compared the new approved 3 mg dose compared to weight-loss in obese Asian NAFLD patients, showing a benefit in body weight reduction and liver enzymes decrease in patients treated with liraglutide, but without significant differences in comparison with lifestyle intervention group [95]. However, it should be underlined that results on efficacy and safety of liraglutide have been obtained in small cohorts of patients with NASH and they need further confirmation in large-scale studies.

SGLT-2 is expressed by epithelial cells of the proximal contorted tubule and it has an important role in promoting glycosuria. SGLT-2 inhibitors reduce glucose reabsorption by kidney, determining a control of blood glucose levels that is independent from secretion and insulin sensitivity, unlike other antidiabetic drugs. Different animal models showed an association between SGLT-2 inhibitors use and improvement of steatosis and fibrosis [96,97]. Interestingly, it has been shown that Tofogliflozin may reduce the risk of hepatocellular carcinoma in obese and diabetic mice models [98]. The impact of SLGT-2 inhibitors on histological outcomes in patients with NASH has not yet been evaluated and only ipraglifozin and empaglifozin have been assessed in RCTs including NAFLD patients. Ipraglifozin was compared with pioglitazone in a RCT conducted on 66 NAFLD patients and type 2 diabetes ones, showing no significant differences in aminotransferase levels, HbA1c and glucose levels [99]. However, it has been demonstrated that only the arm treated with ipraglifozin experienced a significant improvement in visceral fat and body weight. Similarly, compared to standard of care, treatment with empaglifozin was associated with a significant decrease in liver fat, assessed with MRI-PDFF, and with an improvement in ALT levels [100]. Anyway, it should be underlined that SLGT-2 inhibitors have been shown to be associated with side effects, such as urinary and genital infections, that are favored by glycosuria and a potential increase in the risk of breast and bladder cancer. For these reasons, their use in NASH needs further evidence, in terms of both efficacy and safety [101,102].

FGF21, belonging to the FGFs family, a group of signaling proteins involved in cell proliferation and in glucose regulation and lipid metabolism. In particular, FGF21 is mainly produced by the liver and it has a role in the regulation of the adaptive fasting response [103,104]. Pathological conditions characterized by an increase in levels of free fatty acids lead to an upregulation of FGF21, mediated by PPARα, and FGF21 circulating levels are increased in obese animal models [105,106]. It has been shown that pharmacological administration or FGF21 overexpression in obese rodent models is associated with significant metabolic effects on white adipose tissue in terms of reduction of glucose levels, free fatty acids and triglycerides, weight loss, and increased secretion of adiponectin [107,108]. Chronic treatment with FGF21 also improves liver function, through the downregulation of genes involved in hepatic glycolysis, de novo fatty acid synthesis, and triglyceride synthesis, although it is not clear if FGF21 induced improvement in hepatic metabolism is related to a direct action of FGF21 on the liver or not [104,109]. FGF21 acts also on pancreas, protecting β-cells from metabolic stress and lowering the glucagon secretion [110].

Pegbelfermin (BMS-986036) is a pegylated recombinant analogue of FGF21 that was shown to be able to improve NAS and fibrosis in mice models and to improve insulin sensitivity and adiponectin levels in obese patients with or without type 2 diabetes [111,112]. A recent multicenter phase IIa RCT assessed pegbelfermin in non-cirrhotic patients with biopsy-proven NASH, comparing subcutaneous injection of pegbelfermin 10 mg once a day (*n* = 25) with 20 mg dosage once a week (*n* = 24) and subcutaneous placebo (*n* = 26), for 16 weeks [113]. Although no histological outcome was assessed, pegbelfermin showed to be safe and well tolerate and an absolute decrease of 6.8% in MRI-PDFF from baseline was observed in the arm treated with 10 mg once a day, that was significantly higher if compared to placebo (−1.3%. *p* = 0004), regardless of the presence of type 2 diabetes. Although mean reduction in liver stiffness (assessed with MR elastography) was higher in patients who received placebo, the prevalence of patients who obtained a reduction of at least 15% in liver stiffness was significantly higher in patients treated with pegbelfermin 10 mg in comparison with placebo (36% vs. 7%, respectively). Treatment with pegbelfermin resulted in a decrease in aminotransferase levels, increase in adiponectin levels and interestingly, it was associated with a significant reduction in PRO-C3 levels (a biomarker of fibrosis), compared to placebo. No significant reductions in body weight were observed in association with pegbelfermin treatment. Overall, these data suggest that treatment with subcutaneous pegbelfermin is safe and effective in patients with NASH, but studies with larger sample size, longer duration of treatment, and designed for assessment of histological outcomes are needed in the future.

### 3.6. Emricasan

Liver injury and apoptosis are some of the distinctive features that differentiate NASH from simple steatosis. They are related with accumulation of immune cells and cytokines production (i.e., tumor necrosis factor-α (TNF- α)), leading to chronic inflammation and fibrosis [114]. Activation of caspase is critically involved both in intrinsic and in extrinsic apoptotic pathways. Emricasan is an oral irreversible pan-caspase inhibitor, which showed an effect in improvement of aminotransferase levels in chronic hepatitis C and it was also assessed in NAFLD mice models, demonstrating to be able to improve liver inflammation and fibrosis, without effects on hepatic fat accumulation [115,116]. The potential usefulness of Emricasan in NAFLD was assessed in a recent phase II trial including 38 non-cirrhotic patients with elevated aminotransferase levels, randomized to receive emricasan 25 mg twice daily or placebo for 28 days. Treatment with emricasan resulted associated with a significant reduction of ALT and cytokeratin-18 levels, a well-known marker of apoptosis, although no histological outcomes were assessed [117]. A phase IIb trial (ENCORE-NF) was conducted on 318 patients with biopsy-proven NASH evaluating a longer duration of emricasan treatment (72 weeks) with different dosages (5 mg and 50 mg twice a day) in comparison with placebo, and primary outcomes was improvement in fibrosis without NASH worsening. However, definitive results have not yet been published. Interestingly, the impact of emricasan on more advanced stages of liver disease is being evaluated and its effect on portal hypertension was recently assessed in patients with compensated cirrhosis and hepatic vein pressure gradient (HVPG) ≥ 5. An exploratory non-randomized study on 23 patients (56.5% with NASH) showed no significant change in HVPG after emricasan treatment, although a significant HVPG decrease was observed only in patients with severe portal hypertension [118]. Finally, preliminary data from a double-blind placebo controlled randomized trial of emricasan in patients with NASH cirrhosis and severe portal hypertension (HVPG ≥ 12) have been presented during last Internation Liver Congress 2019 [119]. A sample of 263 patients were randomized to receive 5, 25, 50 mg or placebo twice daily for 48 weeks and primary outcome was reduction of HVPG at 24 weeks. Primary endpoint was not met and only the subgroup of patients with baseline HVPG ≥ 16 showed a decrease in HVPG in association with emricasan treatment. However, significant decreases in aminotransferase levels, cytokeratine 18 and caspase 3/7 were observed, suggesting a potential protective effect on liver injury.

### 3.7. Antifibrotic Drugs

In NASH patients, hepatic fibrosis is the main characteristic predicting liver related and all cause (e.g., cardiovascular disease) mortality; the risk of liver-related mortality increases in fibrosis stage [103]. Hepatic macrophages would play an important role in the pathogenesis, development and progression of hepatic fibrosis [120,121]. In cases of acute or chronic injury to the liver, Ly-6Chi monocytes differentiate into proinflammatory macrophages that interact with HSCs in order to promote fibrosis through the TGFβ production. The chemokine (C-C motif) ligand (CCL) 2/C-C chemokine receptor 2 (CCR2) pathway plays a relevant role in hepatic recruitment of Ly-6Chi monocytes. It has also been shown that phagocytosis of cellular debris encourages a phenotypic switch from Ly-6Chi macrophages to Ly-6C low macrophages that are the main source of MMPs and promotes fibrosis resolution [120,121]. A recent study has shown that CCR2+ macrophages increased parallel to NASH severity and fibrosis stage, with a concomitant inflammatory polarization of these cluster of differentiation 68+, portal monocyte-derived macrophages (MoMF) [122].

Accordingly, targeting these cells might represent a promising therapeutic strategy in the management of non-alcoholic fatty liver disease.

Cenicriviroc (CVC) is an oral dual antagonist of C-C motif chemokine receptor CCR types 2 and 5, which prevents macrophage trafficking and efficiently inhibits monocyte infiltration [123]. CCR2 antagonism by CVC is expected to reduce the recruitment, migration, and infiltration of proinflammatory monocytes and macrophages at the site of liver injury [124]. CVC-mediated antagonism of CCR5 is expected to additionally impair the migration, activation, and proliferation of collagen-producing activated hepatic stellate cells/myofibroblasts [125]. Several studies [126,127,128] support the anti-inflammatory and antifibrotic properties belonging to CVC in animal models of liver and kidney fibrosis [125,129]. It also improve non-invasive markers of hepatic fibrosis, as observed in human immunodeficiency-infected subjects and has a favorable safety profile including patients with cirrhosis Child Pugh A-B [128,130]. The efficacy and safety of CVC in patients with NASH and liver fibrosis (LF) were assessed in the CENTAUR (NCT02217475) study [124]. This is a phase 2b, double-blind, randomized, placebo-controlled, multinational study that enrolled 289 subjects with NASH, a non-alcoholic fatty liver disease activity score (NAS) ≥ 4 and LF (NASH Clinical Research Network stage 1–3). Subjects were randomized to receive CVC 150 mg per os once daily (Arm A) or placebo (Arm C). After 1 year, half of the subjects receiving placebo crossed over to CVC (Arm B), based on preplanned randomization, for a second year of treatment. After 1 year, the primary endpoint of NAS improvement in the intent-to-treat population was achieved in a similar proportion of subjects on CVC (*N* = 145) and placebo (*N* = 144; 16% vs. 19% respectively; odds ratio (OR), 0.82 (95% confidence interval, 0.44–1.52); *p* = 0.52). Resolution of steatohepatitis and no worsening of fibrosis, a key secondary outcome, was also observed in similar rates in the 2 groups (8% vs. 6%, respectively; OR, 1.40 (95% CI, 0.54–3.63); *p* = 0.49). The improvement in fibrosis by ≥1 stage and no worsening of steatohepatitis, another key secondary outcome, was observed in more patients on CVC if compared to placebo (20% vs. 10%, respectively; OR 2.20 (95% CI, 1.11–4.35); *p* = 0.02). When the two key secondary endopoints were analyzed together (composite secondary endpoint: ‘complete resolution of SH and no worsening of fibrosis stage’ and ‘improvement in fibrosis stage by 1 stage and no worsening of SH’) subjects receiving CVC had a greater chance to achieve the outcome in comparison to subjects who received placebo (18% vs. 10%; OR, 1.93 (95% CI, 1.04–3.61); *p* = 0.05). Patients with higher NAS, prominent hepatocellular ballooning, higher fibrosis stage, mild or no portal inflammation and with higher body mass index showed greater improvement with CVC treatment. CCL-2 and -4 increased in CVC treated patients, confirming CCR2 and CCR5 blockade by CVC. Safety and tolerability of CVC were comparable to placebo [131]. Finally, twice as many subjects on CVC compared to placebo achieved the clinically important key secondary outcome of improvement in fibrosis by 1 stage and no worsening of SH. The 2 year primary analysis of CENTAUR study show that CVC was well tolerated and provided antifibrotic activity in adults with NASH and LF, confirming the Year 1 primary endpoint: similar proportion on CVC or placebo achieved ≥1-stage fibrosis improvement and no worsening of NASH (15% (15/99) in Arm A vs. 17% (9/54) in Arm C); however, a higher proportion on CVC achieved ≥2-stage fibrosis improvement and no worsening of NASH (11% (7/65) in Arm A vs. 3% (1/34) in Arm C) [132]. Currently AURORA study (NCT03028740), a phase III trial, is ongoing and it will evaluate the effects (efficacy and safety) of CVC in 2000 adult NASH subjects (F2–F3).

Selonsertib (formerly GS-4997) is a selective ASK1 inhibitor. early human studies revealed that selonsertib reduces inflammation and hepatocyte apoptosis [133]. The activation of apoptosis- and oxidative stress-ralated pathways are a hallmark of NASH and fibrosis progression [133]. Apoptosis signal-regulating kinase 1 (ASK1), a serine/threonine kinase, known as mitogenactivated protein kinase 5, leads to phosphorylation of c-Jun N-terminal kinase (JNK) and p38 mitogen-activated kinase (p38 MAPK) that, by hepatic stellate cell activation, mediate pathways leading to apoptosis, fibrosis, lipogenesis and the release of inflammatory cytokines [134]. In preclinical models of NASH, genetic deletion or pharmacological inhibition of ASK1 reduces p38 and JNK phosphorylation, resulting in reduced hepatic steatosis, inflammation, and fibrosis [135]. The efficacy and safety of selonsertib, with and without simtuzumab in NASH patients, was studied in a phase 2, open label, randomized controlled trial (NCT02466516) conducted throughout the USA and Canada [136]. The trial recruited participants with biopsy-proven NASH, with presence of stage F2–F3 fibrosis and NAS ≥ 5. 72 patients were randomly assigned in a 2:2:1:1:1 ratio to receive 24 weeks of treatment with 6 mg or 18 mg of selonsertib alone, 6 mg or 18 mg of selonsertib with 125 mg of simtuzumab, or 125 mg of simtuzumab alone. Because of the lack of effect of simtuzumab on histology or selonsertib pharmacokinetics, selonsertib groups with and without simtuzumab were pooled. Selonsertib is better than placebo in achieving the primary efficacy endpoint of fibrosis improvement of one stage or greater after 24 weeks of treatment. However, phase 3 studies of selonsertib in patients with NASH and bridging fibrosis -STELLAR-3 (NCT03053050)- and compensated cirrhosis -STELLAR-4 (NCT03053063)- definitively failed to demonstrate the efficacy of selonsertib for NASH treatment if compared to placebo.

Simtuzumab (formerly GS-6624) is a humanized monoclonal antibody with an immunoglobulin IgG4 isotype, which is directed against human lysyl oxidase-like molecule 2 (LOXL2). LOXL2 is an enzyme that promotes the cross-linking of collagen and elastin, therefore leading to remodeling of the extracellular matrix representing a key component in the core regulatory pathway of fibrogenesis [137]. Preclinical analysis from a murine model of advanced fibrosis suggested that simtuzumab shows an additive effect when combined with an ASK1 inhibitor [138,139]. However, two phase 2b trials of patients with bridging fibrosis or compensated cirrhosis due to NASH, showed that simtuzumab was not effective in decreasing hepatic collagen content or hepatic venous pressure gradient.

## 4. Conclusions

In the next few years the pandemic of NAFLD will lead to an increase of NASH patients and those with fibrosis at higher risk of developing liver-related complications. This picture highlights the need for an effective pharmacological therapy. A number of drugs targeting different pathogenetic pathways are under investigation, even if only few of them already confirmed their efficacy in phase III trials, some other definitively failed to prove their superiority respect to placebo, and some other are still under investigation (Table 1). In conclusion, the challenge for the next future is a correct approach to NASH therapy that will take into account lifestyle correction—the backbone of NASH therapy, control of associated metabolic disorders, and, when available, a rational use of specific drug(s) with potentially different target according to the phenotype of the NASH patient.

## Figures and Tables

**Figure 1 ijerph-16-04334-f001:**
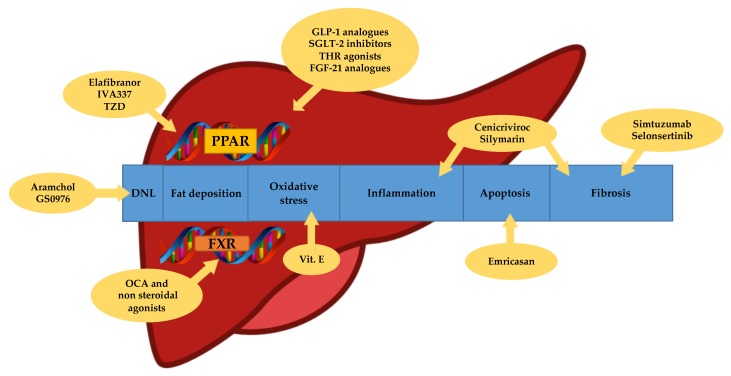
Targets of NASH therapy. DNL: de novo lipogenesis. TZD: thiazolidinediones. OCA: obethicolic acid. PPAR: peroxisome proliferator-activated receptors. FXR: farnesoid X receptors. GLP-1: glucagon-like peptide 1. SGLT-2: sodium-glucose transport protein 2. THR: thyroid hormone receptor. FGF-21: fibroblast growth factor 21.

**Table 1 ijerph-16-04334-t001:** Summary of the analyzed drugs, their mechanisms of action and scientific evidences.

Drugs	Mechanisms of Action	Scientific Evidence
Thiazolidinediones	- Modulate adipose tissue distribution, decreasing visceral fat; - Regulate insulin sensitivity by the activation of peroxisome proliferator-activated receptor-γ	- Pioglitazone group (58%) achieved primary outcome (NAS impovement ≥ 2 points without worsening of fibrosis) than patients in the placebo group (17%) [9].
Vitamin E	- Anti-oxidant effect that can protect cellular structure integrity against injury from lipid peroxidation and oxygen-free radicals	- Vitamin E use was associated with improved clinical outcomes in patients with histological proven NASH and bridging fibrosis or cirrhosis [13]. - Vitamin E treatment, as compared with placebo, was associated with a significantly higher rate of improvement in NASH (43% vs. 19%, *p* = 0.001), but not significant if pioglitazone as compared with placebo (34% and 19%, *p* = 0.04) [14]. - Vitamin E and pioglitazone were associated with reductions in hepatic steatosis (*p* = 0.005 and *p* < 0.001, respectively) and lobular inflammation (*p* = 0.02 and *p* = 0.004, respectively) but not with improvement in fibrosis scores (*p* = 0.24 and *p* = 0.12 respectively) [14].
Silymarin	- Antioxidant, anti-inflammatory, and antifibrotic properties	- Silymarin (700 mg 3 times daily for 48 weeks) determined an histological and non-invasive (based on liver stiffness measurements) improvement of fibrosis despite placebo group (22.4% and 6% respectively, *p* = 023) [25].
Obeticholic acid	- Increase the metabolic rate and reverse high fat-induced diabetes, decreasing adiposity	- 45% patients in the obeticholic acid group had improved liver histology (2 point or greater improvement in NAFLD activity score without worsening of fibrosis) compared with 21% of patients in the placebo group (*p* = 0.0002) [35]; - Improvement of NASH without worsening of fibrosis was not statistically significant. Itch was the most common adverse event and forced the discontinuation especially in obeticholic acid 25 mg arm. Increases in LDL were also observed by week 4 [37].
NGM282	- Non-tumorigenic, engineered variant of fibroblast growth factor-19 that binds FGF receptors 1c and 4 to reduce lipotoxicity due to lipid accumulation	- Induced marked relative reduction from baseline in steatosis with a 57% and 45% (*p* < 0.0001) at the 6- and 3-mg doses, respectively. Improvements in ALT (*p* < 0.0001), AST (*p* < 0.0001) were also observed, but not in glycemic control. Injection site reactions were reported in 54% and 41% of subjects in the 6- and 3-mg dose groups relative to 7% in the PBO group [40].
Cilofexor	- Non-steroidal FXR agonist	- Phase II study with 140 NASH patients, treated with cilofexor at 100 or 30 mg or placebo orally, once daily for 24 weeks. A significant decrease in hepatic steatosis of at least 30% assessed by MRI-PDFF was observed in 38.9% of patients treated with cilofexor 100 mg (*p* = 0.011), 14% treated with cilofexor 30 mg (*p* = 0.87), and 12.5% treated with placebo. Further significant observations in the cilofexor-treated patients were improvements in serum GGT and serum 7α-hydroxy-4-cholesten-3-one (C4) [42].
Tropifexor	- Non-steroidal FXR agonist	- At 12 weeks, a significant decrease in hepatic steatosis of at least 5% assessed by MRI-PDFF was observed in 33.3% of patients treated with Tropifexor 90 μg, 27.8% treated with Tropifexor 60 μg, and 14.6% treated with PBO. Furthermore, a dose–response decrease in GGT levels was observed as well as increases in FGF19. Adverse events, especially pruritus and increase in LDLc and HDLc, were reported in 90 μg Tropifexor arm [45].
Elafibranor	- Playing a key role in glucose, lipid metabolisms and inflammation; - Improvement of insulin resistance	- With the intention to treat analysis, 92 patients received placebo and 93 and 89 patients received Elafibranor 80 and 120 mg, respectively. Outcome: a score of 0 of at least 1 between steatosis, ballooning and inflammation (reversal of NASH) without worsening of fibrosis (progression of patients with fibrosis stage ≤ 2 to F3-F4 or from F3 to F4). no differences were observed between patients treated with Elafibranor and placebo (OR 1.53, 95% CI 0.70−3.34, *p* = 0.280); conversely, using the modified definition, the response rate was significantly higher in those who received Elafibranor 120 mg in comparison with placebo (OR 2.31, 95% CI 1.02−5.24, *p* = 0.045), with a lower placebo effect [67]; - Post-hoc analysis restricted to 234 patients with NAS score ≥ 4 at baseline showed that treatment with Elafibranor 120 mg (*n* = 75) was significantly associated with higher response rate in comparison with placebo (*n* = 76), using both definitions of response, although it should be considered that no stratification according this covariate was performed at randomization [67].
Lanifibranor	- Playing a key role in glucose, lipid metabolisms and inflammation; - Improvement of insulin resistance	- Demonstrated antifibrotic efficacy in the CCl4-induced liver fibrosis model in terms of prevention or reversion of fibrosis [68].
GS-0976	- Inhibitor of de novo lipogenesis	- A group of 93 patients with NASH (diagnosed with magnetic resonance elastography or with histology) were assigned to GS-0976 (46 patients to 20 mg dosage and 47 to 5 mg), while 26 patients received placebo for 12 weeks. In comparison with placebo, GS-0976 20 mg was significantly associated with a relative reduction of at least 30% from baseline in magnetic resonance imaging-estimated MRI-PDFF (48% vs. 15%, *p* = 0.004), as well as median relative decreases in MRI-PDFF were more pronounced for GS-0976 20 mg in comparison with placebo [72,73].
Aramchol	- Inhibitor of de novo lipogenesis; - Inhibition of steatoyl-coenzime A desaturase	- A group of 247 patients with histologically proven NASH, NAS ≥ 4 without cirrhosis and type 2 diabetes/prediabetes were randomized to receive Aramchol 400 mg (*n* = 101), Aramchol 600 mg (*n* = 98) and placebo (*n* = 48). Primary endpoint was the decrease from baseline in steatosis assessed by MRS; secondary endpoints were fibrosis score improvement without worsening of NASH and NASH resolution without worsening of fibrosis. Treatment with Aramchol 600 mg was associated with reduction of at least 5% in liver fat from baseline in 47% of patients (versus 24% in placebo, *p* = 0.0279, OR 2.77 (95% CI 1.12–6.89) and with NASH resolution alone in 19% of patients (versus 7.5% in placebo, *p* = 0.046) [77].
Resmetirom	- Highly selective THR β agonist that showed positive effects on reduction of LDL cholesterol and triglycerides	- Dosage of 80 mg once daily was assigned to 78 patients and 38 patients received placebo for 36 weeks. Primary endpoint at 12 weeks (relative reduction in MRI-PDFF) was met (−36% in treated arm versus −9% in placebo arm). Conversely, histological endpoint at 36 weeks (2-points NAS reduction with ≥ 1-point decrease in ballooning or inflammation) was not met (51% in treated arm vs. 32% in placebo arm, *p* = 0.09). However, limiting the analysis to patients treated with Resmetirom who obtained a reduction of at least 30% in MRI-PDFF, response rate was significantly higher (65%, *p* = 0.006) in comparison with placebo, suggesting a relationship between early reduction in liver fat at 12 weeks and NASH histological improvement at 36 weeks [85].
VK-2809	- THR β agonist	- A 12-week phase II RCT, in whom 16 patients received VK-2809 10 mg once a day, 16 patients received 10 mg every other day and 15 patients were assigned to placebo. Daily dose administration was associated with a median reduction of 60% (*p* < 0.01) at 12 weeks of liver fat content in comparison with placebo, although an increase in ALT levels was observed, especially during early treatment. However, after 12 weeks of treatment, ALT levels were similar between VK-2809 and placebo [86].
Liraglutide	- Long-acting GLP-1 analogue	- It was associated with a significant improvement in liver enzymes and with a good safety profile in an individual patient data meta-analysis of six randomized controlled trials, concerning more than 4000 diabetic patients [91]; - A treatment of 48 weeks of subcutaneous liraglutide (1.8 mg/day) were compared with placebo and primary endpoint was NASH resolution without impairment of fibrosis. Thirty-nine percent of patients treated with liraglutide compared to 9% placebo group patient (*p* = 0.019) met the primary endpoint, whereas no significant differences were observed in progression of fibrosis between two groups [92].
Ipraglifozin	- SGLT-2 inhibitor	- The arm treated with ipraglifozin experienced a significant improvement in visceral fat and body weight [98].
Pegbelfermin	- Pegylated recombinant analogue of FGF21	- Although no histological outcome was assessed, pegbelfermin showed to be safe and well tolerate and an absolute decrease of 6.8% in MRI-PDFF from baseline was observed in the arm treated with 10 mg once a day, that was significantly higher if compared to placebo (−1.3%. *p* = 0004), regardless of the presence of type 2 diabetes. Although mean reduction in liver stiffness (assessed with MR elastography) was higher in patients who received placebo, the prevalence of patients who obtained a reduction of at least 15% in liver stiffness was significantly higher in patients treated with pegbelfermin 10 mg in comparison with placebo (36% vs. 7%, respectively) [112].
Emricasan	- Oral irreversible pan-caspase inhibitor	- Treatment with emricasan resulted associated with a significant reduction of ALT and cytokeratin-18 levels, a well-known marker of apoptosis, although no histological outcomes were assessed [116]. - No significant change in HVPG after emricasan treatment, although a significant HVPG decrease was observed only in patients with severe portal hypertension [117].
Cenicriviroc	- Oral dual antagonist of C-C motif chemokine receptor types 2 and 5	- Dosage of 150 mg per os once daily. After 1 year the primary endpoint of NAS improvement in the intent-to-treat population was achieved in a similar proportion of subjects on CVC (*N* = 145) and placebo (*N* = 144; 16% vs. 19% respectively; odds ratio (OR), 0.82 (95% confidence interval, 0.44–1.52); *p* = 0.52). Resolution of steatohepatitis and no worsening of fibrosis, a key secondary outcome, was also observed in similar rates in the 2 groups (8% vs. 6%, respectively; OR, 1.40 (95% CI, 0.54–3.63); *p* = 0.49). The improvement in fibrosis by ≥1 stage and no worsening of steatohepatitis, another key secondary outcome, was observed in more patients on CVC if compared to placebo (20% vs. 10%, respectively; OR 2.20 (95% CI, 1.11–4.35); *p* = 0.02) [130].
Selonsertinib	- Selective apoptosis signal-regulating kinase 1 inhibitor	- NASH patients, with presence of stage F2-F3 fibrosis and NAS ≥ 5. 72 patients were randomly assigned in a 2:2:1:1:1 ratio to receive 24 weeks of treatment with 6 mg or 18 mg of selonsertib alone, 6 mg or 18 mg of selonsertib with 125 mg of simtuzumab, or 125 mg of simtuzumab alone. Because of the lack of effect of simtuzumab on histology or selonsertib pharmacokinetics, selonsertib groups with and without simtuzumab were pooled. Selonsertib is better than placebo in achieving the primary efficacy endpoint of fibrosis improvement of one stage or greater after 24 weeks of treatment [136].
Simtuzumab	- Humanized monoclonal antibody with an immunoglobulin IgG4 isotype, which is directed against human lysyl oxidase-like molecule 2	- Peclinical analysis from a murine model of advanced fibrosis suggested that simtuzumab shows an additive effect when combined with an ASK1 inhibitor [138].

NAS: NAFLD activity score; NASH: non-alcoholic steatohepatitis; NAFLD: non-alcoholic fatty liver disease; LDL: low density lipoproteins; ALT: alanine aminotransferase; AST: aspartate aminotransferase; PBO: placebo; FXR: farnesoid X receptor; MRI-PDFF: Magnetic Resonance Imaging Proton Density Fat Fraction; GGT: gamma glutamyl transpeptidase; FGF: fibroblast growth factor; HDL: high density lipoproteins; THR: thyroid hormone receptor; GLP: glucagon like peptide; SGLT: sodium-glucose transporter.

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
