# Peer review of "Pharmacological Therapy of Non-Alcoholic Fatty Liver Disease: What Drugs Are Available Now and Future Perspectives"

_ijerph, 2019, doi:10.3390/ijerph16224334_

Round 1

Reviewer 1 Report

The review of Pennisi et al. gives a comprehensive and interesting overview about the pharmacological therapy approaches of NAFLD, in particular, of available drugs and drugs that are in development. It is well done, molecular mechanism of the different drugs were described and explained as well as current studies were presented. However there are some issues that need to be addressed.

Major comments

English language needs improvement and major edit. There are several spelling mistakes (for example sylimarin instead of silymarin, hystological instead of histological etc.), grammar mistakes and single words are missing, therefore it is hard to read the text and to focus on the content of the manuscript. Aim of the review of the abstract section is not clear, and too general. The focus is on pharmacological therapy of NAFLD. This should be pointed out. In Figure 1, it is not clear where GLP-1 analogues, SGLT-2 inhibitors are function or which effects they have on liver disease. Subheading "Available drugs" is misleading or too general. In section 2, therapeutic approaches are mentioned based on guidelines, this should be make clear. Define low plasma level of vitamin E (lines 91,92) and how long could vitamin E used based on current evidence (line 113)? Shorten chapter 3.2 PPAR agonists, especially the section to  elafibranor, chapter 3.7 antifibrotic drugs, especially the section to cenicriviroc. A table that summarizes all drugs, their underlying mechanisms, results of studies of different phases, used concentration etc might be an useful addition to the text. Author Contributions, Funding, and COI is missing.

Minor comments

Please use abbreviations in a consistent manner through the manuscript, e.g. NAFLD, NASH, HCC etc. Please check upper and low case for some words. In chapter 3.7 output style of references is not consistent.

Author Response

Reviewer 1

Comments and Suggestions for Authors

The review of Pennisi et al. gives a comprehensive and interesting overview about the pharmacological therapy approaches of NAFLD, in particular, of available drugs and drugs that are in development. It is well done, molecular mechanism of the different drugs were described and explained as well as current studies were presented. However there are some issues that need to be addressed.

Major comments

English language needs improvement and major edit. There are several spelling mistakes (for example sylimarin instead of silymarin, hystological instead of histological etc.), grammar mistakes and single words are missing, therefore it is hard to read the text and to focus on the content of the manuscript.

Aim of the review of the abstract section is not clear, and too general. The focus is on pharmacological therapy of NAFLD. This should be pointed out.

In Figure 1, it is not clear where GLP-1 analogues, SGLT-2 inhibitors are function or which effects they have on liver disease.

Subheading "Available drugs" is misleading or too general.

In section 2, therapeutic approaches are mentioned based on guidelines, this should be make clear.

Define low plasma level of vitamin E (lines 91,92) and how long could vitamin E used based on current evidence (line 113)?

Shorten chapter 3.2 PPAR agonists, especially the section to  elafibranor, chapter 3.7 antifibrotic drugs, especially the section to cenicriviroc.

A table that summarizes all drugs, their underlying mechanisms, results of studies of different phases, used concentration etc might be an useful addition to the text.

Author Contributions, Funding, and COI is missing.

Minor comments

Please use abbreviations in a consistent manner through the manuscript, e.g. NAFLD, NASH, HCC etc. Please check upper and low case for some words.

In chapter 3.7 output style of references is not consistent.

Response to reviewer 1: we thank the Reviewer for her/his comments.

We reviewed the English style as suggested. Surely, we tried to modify as suggested the abstract section regarding the aim of the review bearing in mind the word count limit of the Journal. Surely, we made a mistake in the orientation of the arrow. We modified the figure. Because of it is a summary figure its role is not to give precise indication to the reader regarding the mechanisms of action of all the pharmacological strategy indicated. We think that a precise explanation figure could be extremely big and complex to read and understand. We modified as suggested the title. As suggested we specified as indicated. As suggested we added the missed information. We modified as suggested the section indicated. As suggested we designed a summary table with the information required. We added the missed information. We corrected the mistakes. We corrected the mistakes.

Reviewer 2 Report

This paper overviewed pharmacological treatment for NASH/NAFLD and described currently available drugs and those in development.

Although the topic is very attractive, each section is too descriptive, and I am afraid the explanation of each drug may not be well balanced; some are so brief, and others are too redundant. Overall, the general perspective of NASH treatment should be provided. As for clinical studies, the ethnicity of the subjects included is critical, since there are many differences in clinical features of NASH among populations, especially between the East and the West. It should be also mentioned that one of the challenges of drug development for NASH is the lack of consistent diagnostic methods without liver biopsy. There are several grammatical errors throughout the paper (for example, “gived” in line 81), which significantly reduced the reliability of the paper.

Some other specific points are as follows.

Fig1: Several mechanisms of NASH development were lined in tandem, but this may be misleading because these do not happen sequentially. Insulin sensitizers (TZD): The relationship between insulin resistance and NASH is essential but complicated, since lipogenic effects of insulin may not always be impaired. In this sense, the resolution of NASH by TZD is not so straightforward, and the risk of weight gain should be more emphasized. (NGM-282): If NGM-282 is a variant of FGF19, them it should not be listed in the FXR agonist section. (THRβ agonist): If I remember correctly, THRβ agonists can exert specific effects on liver because THRβ has a distinct tissue distribution. Please make clear the feasibility of using such agonists. (GLP-1 analogues) and (SGLT-2 inhibitors): These should be listed in clinically available drugs rather than drugs in development.

Author Response

Reviewer 2

Comments and Suggestions for Authors

This paper overviewed pharmacological treatment for NASH/NAFLD and described currently available drugs and those in development.

Although the topic is very attractive, each section is too descriptive, and I am afraid the explanation of each drug may not be well balanced; some are so brief, and others are too redundant. Overall, the general perspective of NASH treatment should be provided. As for clinical studies, the ethnicity of the subjects included is critical, since there are many differences in clinical features of NASH among populations, especially between the East and the West. It should be also mentioned that one of the challenges of drug development for NASH is the lack of consistent diagnostic methods without liver biopsy. There are several grammatical errors throughout the paper (for example, “gived” in line 81), which significantly reduced the reliability of the paper.

Some other specific points are as follows.

Fig1: Several mechanisms of NASH development were lined in tandem, but this may be misleading because these do not happen sequentially. Insulin sensitizers (TZD): The relationship between insulin resistance and NASH is essential but complicated, since lipogenic effects of insulin may not always be impaired. In this sense, the resolution of NASH by TZD is not so straightforward, and the risk of weight gain should be more emphasized. (NGM-282): If NGM-282 is a variant of FGF19, them it should not be listed in the FXR agonist section. (THRβ agonist): If I remember correctly, THRβ agonists can exert specific effects on liver because THRβ has a distinct tissue distribution. Please make clear the feasibility of using such agonists. (GLP-1 analogues) and (SGLT-2 inhibitors): These should be listed in clinically available drugs rather than drugs in development.

Response to reviewer 2: we thank the Reviewer for her/his comments.

We redistributed the length of each section properly giving to the paper an homogeneous distribution. Moreover we improved the general readability of the manuscript. We understand the request of the reviewer and we are completely agree. Unfortunately, as suggested by the title of the manuscript, this is not an objective of the current review, also because at the moment a precise, scientifically accepted and well codified drug therapy for NASH doesn’t exist. We added some sentences highlighting the ethnicity differences in NAFLD setting. We inserted the information in the main text, in the introduction section. We reviewed the English style as suggested. We modified the figure as suggested eliminating the confounding factors. We not declared the role of TZD in the resolution of NAFLD/NASH just because the relationship between insulin resistance and the disease is very complex and beyond the rationale of this manuscript. We have limited our attention on the available data from clinical studies indicated properly in the reference section. We repositioned properly the indicated section. As highlighted in the main text at the moment there isn’t any medication of this category approved for the clinical use in this setting. As highlighted in the main text at the moment there isn’t any medication of this category approved for the clinical use in this setting. We decided to wrote about the drugs in development in order to answer to one of the scopes of the current paper.

Reviewer 3 Report

The manuscript from Grazia et al. is comprehensive and focused on pharmacological therapies available for NAFLD. My minor comments are as follows:

There are many small paragraphs in the introduction section which can be combined accordingly to make two to three maximum paragraphs.  Authors can also discuss recent study (J. Hepatol. 2019, 71, 793–801) to show even higher prevalence of NAFLD and NASH among T2D patients. Authors need to be consistent while using abbreviations such as NAFL, NASH, NAS, LF as this may be misleading. NAFL and NAS may be the same stage as LF and NASH for general audience.  Line 647, 648 and 649: Please note that these sections are left blank. There are few typos in the text so need to check thoroughly, such as FGF2 in the line 472.  

Author Response

Reviewer 3

Comments and Suggestions for Authors

The manuscript from Grazia et al. is comprehensive and focused on pharmacological therapies available for NAFLD. My minor comments are as follows:

There are many small paragraphs in the introduction section which can be combined accordingly to make two to three maximum paragraphs.  Authors can also discuss recent study (J. Hepatol. 2019, 71, 793–801) to show even higher prevalence of NAFLD and NASH among T2D patients. Authors need to be consistent while using abbreviations such as NAFL, NASH, NAS, LF as this may be misleading. NAFL and NAS may be the same stage as LF and NASH for general audience.  Line 647, 648 and 649: Please note that these sections are left blank. There are few typos in the text so need to check thoroughly, such as FGF2 in the line 472. 

Response to reviewer 3: we thank the Reviewer for her/his comments.

In accordance to the suggestion we reduced as maximum as possible the number of the paragraphs. We added the paper in the specific section. We corrected the identified mistake in the main text. We added the missed information. We corrected the mistakes.

Round 2

Reviewer 1 Report

The authors have considerably improved the manuscript according to my comments and suggestions.

Author Response

Reviewer 1

Comments and Suggestions for Authors

The authors have considerably improved the manuscript according to my comments and suggestions.

Response:

We thank the reviewer for her/his suggestions.

Reviewer 2 Report

The manuscript was modified according to the suggestions by the reviewers including me, and has been improved significantly.

As for my previous point 9 (the tissue specificity of THRβ), it was not answered clearly yet. I hope it will be briefly mentioned that THRβ is the predominant THR isoform in the liver.

Author Response

Reviewer 2

Comments and Suggestions for Authors

The manuscript was modified according to the suggestions by the reviewers including me, and has been improved significantly.

As for my previous point 9 (the tissue specificity of THRβ), it was not answered clearly yet. I hope it will be briefly mentioned that THRβ is the predominant THR isoform in the liver.

Response:

We clarified as suggested that THRβ is the predominant THR isoform in the liver.